# COUNTERFACTUAL TECHNIQUES FOR ENHANCING CUSTOMER RETENTION

## ABSTRACT

In this paper, we introduce a novel counterfactual reasoning method using eBERT embeddings to convert customers from an e-commerce company who frequently add items to their cart but don't proceed to checkout. We demonstrate that our method i) outperforms existing techniques such as DiCE, GANs, and CFRL in key metrics such as coverage, while also maintaining a low latency; ii) balances high coverage and low latency by adjusting the number of nearest unlike neighbors, highlighting a trade-off between these competing goals; and iii) allows customization of mutable features, improving the practical applicability of our counterfactual explanations.

## 1 INTRODUCTION

In various industries, counterfactual reasoning can be used to analyze 'what-if' scenarios to better understand customer behavior. For instance, customers often add items to their cart but do not proceed to checkout. Similarly, it is also important for loan companies Zhang et al. (2023); Grath et al. (2023), insurance providers Rye & Boyd (2022); Kumar & Ravi (Year), and fraud detection Whitrow et al. (2009); Ngai et al. (2011) companies to explain to their customers why they were denied loans or insurance. Understanding the reasons behind these behaviors and finding strategies to convert these customers is important for improving conversion rates. In recent years, the demand for explainable AI has grown, becoming essential across sectors to ensure transparency and trust among customers Gohel et al. (2023); Verma et al. (2020); Adadi & Berrada (2018).

Existing counterfactual reasoning methods like Nearest Instance Counterfactual Explanations (NICE) Brughmans et al. (2022), Diverse Counterfactual Explanations (DiCE) Mothilal et al. (2019), Generative Adversarial Networks (GANs) Goodfellow et al. (2014), Counterfactual GANs (CounteRGAN) Nemirovsky et al. (2021), and Counterfactuals with Reinforcement Learning (CFRL) Samoilescu et al. (2021) often suffer from either high latency, lack of mutability (i.e., changing only user-specified features), or suboptimal performance in terms of plausibility (i.e., how closely the counterfactual resembles the real world data) and distance (i.e., distance between the original instance and the counterfactual). Providing counterfactuals with actionable and customizable (i.e., mutable) features is essential for achieving business goals.

To address these challenges, we propose a novel counterfactual reasoning method that is built on the foundations of NICE but adds support for mutability and uses contextual embedding techniques to find the nearest neighbors from the opposite class. Our approach involves converting each data row into text, considering both feature and value, and then generating embeddings using a BERT Devlin et al. (2018) model fine-tuned on the company's product titles (eBERT). Through BERT-based embeddings, feature values are represented better semantically, making it an effective method of finding neighbors who are similar but have a different label, increasing plausibility.

Our new technique is particularly valuable in production systems because it supports mutability, allowing business users to specify which features can be changed. As a result, the generated counterfactuals are more actionable. Also, by using embeddings for similarity, the counterfactuals are highly plausible, providing the best plausibility and coverage compared to all the baseline methods.

## 1.1 CONTRIBUTIONS

The primary objective of this study is to develop a novel counterfactual reasoning method to enhance customer retention in e-commerce settings. Our method aims to provide explainable, actionable insights that help identify key factors influencing customers' decision-making processes, specifically for scenarios where customers abandon their carts before completing a purchase. By leveraging eBERT embeddings and supporting mutability, we have achieved the following contributions:

- Developed a new counterfactual reasoning algorithm that improves upon current state-of-the-art methods, such as NICE, DiCE, and GAN-based approaches, by balancing coverage, latency, and plausibility of counterfactuals.

- Introduced an embedding-based approach using eBERT to generate highly plausible counterfactuals that more accurately reflect customer behavior in an e-commerce setting.

- Ensured that the proposed method supports customizable and mutable features, allowing business stakeholders to specify which factors can be realistically adjusted to achieve desired outcomes, thus enhancing the practical applicability of counterfactual explanations.

- Optimized the system for real-time deployment, with a focus on maintaining low latency and high scalability in real-world applications such as customer retention for e-commerce.

- Evaluated the effectiveness of the proposed method across various scenarios, comparing it against existing techniques using key metrics such as coverage, reconstruction error, and L1 distance, and demonstrating its applicability in real-world datasets.

This work aims to address the current limitations in counterfactual reasoning, offering a comprehensive and actionable solution for improving customer retention through explainable AI techniques.

## 2 DATA PREPROCESSING

Our company's dataset consists of 200,000 shopping sessions to understand customer behavior. The dataset includes a total of 47 features extracted primarily from User, Cart, and Listing tables. Of these features, 36 are categorical, such as product categories and purchase data, while the remaining 11 are numerical variables, like shipping costs, item prices, and feedback scores. The task is a binary classification of shopping session outcomes, either 1 (i.e., success) if the customer successfully checked out or 0 (i.e., failure) otherwise.

To preprocess the data for embedding generation, we scaled the price and shipping fee features between 0 and 100 and categorized them into four buckets. The price features were divided into four buckets: 0-20 (*budget*), 20-40 (*affordable*), 40-60 (*premium*), and 60-100 (*luxury*). Shipping fees were also categorized into four buckets: 0-25 (*low*), 25-50 (*medium*), 50-75 (*high*), and 75-100 (*very high*).

For building the classification pipeline, we removed null values, scaled the data using a standard scaler, and encoded categorical data using a binary encoder. We explored several encoding methods for the categorical features, including label encoding, one-hot encoding, target encoding, and binary encoding. While one-hot encoding achieved slightly higher accuracy, it led to overfitting due to high cardinality, increasing the dataset to over 9,000 features and significantly increasing computation time. Binary encoding, although slightly less accurate, reduced dimensionality to 165 features, preventing overfitting and significantly reducing the computation time. Binary encoding was chosen due to its ability to reduce dimensionality and computation time, while preventing overfitting.

The dataset was divided into a training set of 160,000 instances and a test set of 40,000 instances to evaluate model performance. This preprocessing step ensured that both categorical and numerical features were ready for embedding generation.

## 3 CLASSIFICATION

Counterfactual reasoning algorithms incorporate classifiers by using their predictions as feedback during the generation process. Each counterfactual is tested to see if it successfully flips the classifier's original decision or not. We implemented four different types of classifiers—a Random Forest

Table 1: Performance Metrics of Various Classifiers

| METRIC | RANDOM FOREST | MLP | LOG REG | XGBOOST |
|---|---|---|---|---|
| Accuracy | **89.0** | 82.1 | 66.41 | 84.7 |
| Precision | **88.7** | 82.4 | 66.42 | 81.4 |
| Recall | 89.3 | 82.1 | 66.4 | **89.7** |
| F1 Score | **89.0** | 82.1 | 66.4 | 85.4 |

(RF), Logistic Regression, XGBoost Chen & Guestrin (2016), and Multilayer Perceptron (MLP). The Random Forest classifier showed promising results and was used for generating counterfactual explanations. With an F1 score of 89.1% (Table 1), it demonstrated its ability to properly classify and understand customer behavior. The MLP, Logistic Regression, and XGBoost achieved lower accuracy than the Random Forest.

## 4 BASELINE COUNTERFACTUAL METHODS

Here, we describe four strong baseline counterfactual reasoning methods we compared against.

*Diverse Counterfactual Explanations (DiCE)* DiCE Mothilal et al. (2019) primarily focuses on generating feasible and diverse counterfactuals. It extends counterfactual explanations by incorporating determinantal point processes (DPP) Kulesza et al. (2012), which is a probabilistic model used to ensure diversity in the generated examples. This allows DiCE to provide a range of alternatives for changing outcomes.

DPP selects a subset of diverse examples by maximizing the determinant of a kernel matrix built from the examples. This diversity is balanced against proximity, which measures the closeness of counterfactuals to the original input. The method optimizes a loss function that combines y-loss (the difference in prediction), proximity, and diversity, adjusted using hyperparameters $\lambda_1$ and $\lambda_2$. The counterfactuals are generated through gradient descent, which iteratively adjusts feature values to meet the objective while respecting any real-world constraints on feature manipulation.

While DiCE focuses on generating diverse counterfactuals, it suffers from lower coverage and higher reconstruction error. In contrast, our method achieves higher coverage and lower L1 distance, providing more actionable and plausible counterfactuals, especially in e-commerce applications where proximity is more critical than diversity.

*Nearest Instance Counterfactual Explanations (NICE)* NICE Brughmans et al. (2022) generates counterfactual explanations using a nearest unlike neighbor-based approach. The algorithm identifies the nearest neighbor with a different class label and changes one feature value at a time from the original instance to match that of the neighbor. This process generates hybrid instances, guided by a reward function that prioritizes sparsity, proximity, or plausibility, depending on the specific NICE variant.

NICE allows for different objective functions depending on the type of optimization desired. This approach has been tested across various datasets and domains, demonstrating its flexibility and ability to produce minimal yet effective feature changes.

Although NICE produces counterfactuals with minimal L1 distance, it lacks support for mutability, limiting its practical applicability. Our method not only supports feature mutability, making it more customizable for real-world business needs, but also provides faster counterfactual generation with reduced latency.

*Generative Adversarial Networks (Standard GAN & CounteRGAN)* CounteRGAN Nemirovsky et al. (2021) is an extension of Residual GAN (RGAN), designed to generate realistic and actionable counterfactuals by applying small perturbations to existing data points rather than creating new instances from scratch. The idea is to generate subtle modifications that can flip a model's prediction, while ensuring that the changes are realistic and feasible.

In RGAN, the generator produces perturbations that modify input data, while the discriminator attempts to distinguish between real and modified data. CounteRGAN adds a target classifier to this process, which ensures that the generated counterfactuals not only resemble real data but also result in the desired class change. This approach integrates the adversarial learning process with a counterfactual search for more purposeful outcomes.

Standard GAN and CounteRGAN generate counterfactuals with the lowest latency, but at the cost of higher reconstruction error and L1 distance, which reduces plausibility. Our method, while slightly slower, strikes a balance by providing highly plausible counterfactuals with lower L1 distance and full coverage, making it more effective in producing realistic and actionable outcomes.

*Counterfactuals using Reinforcement Learning (CFRL)* CFRL Samoilescu et al. (2021) uses a reinforcement learning framework for counterfactual generation, transforming the optimization process into a learnable task. It enables the generation of multiple counterfactuals in a single forward pass, relying solely on the feedback from the classifier's predictions. This model-agnostic method allows for feature-level constraints, ensuring real-world feasibility.

CFRL uses a critic to estimate rewards from the environment and an actor to output counterfactual latent representations. This method enables high flexibility, as feature-level constraints like immutability can be incorporated via conditioning vectors, ensuring that the generated counterfactuals are plausible and actionable.

CFRL supports mutable feature customization. However, it has the lowest coverage amongst all the other methods and a higher L1 distance compared to our method. By offering a better balance between feature mutability, coverage, and proximity, our method produces more actionable and realistic counterfactuals.

Other counterfactual generation methods such as CERT Sharma et al. (2020) and MACE Karimi et al. (2020) were not considered due to their high latency Schleich et al. (2023), and GeCo Schleich et al. (2023), although it offers many customization options for the counterfactual, suffers from very low coverage Brughmans et al. (2022).

## 5 OUR METHOD

Our novel embedding-based counterfactual reasoning method is designed to address the limitations of NICE by using embeddings to find the nearest unlike neighbors, and it supports mutability. Our method performs better than all other methods supporting mutability, specifically DiCE and CFRL, across all metrics. It also provides better plausibility and faster results compared to NICE. We use eBERT to generate embeddings that capture deep semantic relationships within our data, in order to identify more plausible and actionable counterfactuals.

### 5.1 EMBEDDING GENERATION USING EBERT

In our method, each data sample is transformed into a text representation to generate embeddings for counterfactual reasoning. This transformation involves converting 63 feature names and their respective values into a textual format. For example, a data row with the following attributes:

- `PRICE = affordable`
- `SHPNG_COST = low`
- `PAYMNT_TYPE = CreditCard`
- `QTY_SOLD = 2`

would be represented as: "PRICE *affordable* SHPNG_COST *low* PAYMNT_TYPE *CreditCard* QTY_SOLD *2*"

This text is then standardized to lowercase to maintain consistency before processing. The preprocessed text for each of the 63 features used in our dataset is input into the eBERT model. The eBERT model outputs a 768-dimensional embedding for each data sample, capturing the semantic relationships of the data.

These embeddings serve as the basis for the next step in our method, which involves identifying the nearest unlike neighbors. This process is crucial for generating plausible and contextually relevant counterfactual explanations by calculating the distances between the generated embeddings.

## 5.2 NEAREST UNLIKE NEIGHBORS GENERATION

After generating embeddings using eBERT, the next step involves finding the nearest unlike neighbors, which are used to find a counterfactual explanation. Instead of using the Heterogeneous Euclidean-Overlap Metric as in NICE, our method employs FAISS (Facebook AI Similarity Search) Facebook Engineering (2017) IndexFlatL2 to identify the $k$ nearest unlike neighbors based on the L2 distance.

FAISS is a library optimized for fast similarity searches, particularly for high-dimensional vectors such as embeddings. We use the IndexFlatL2 index type, which is a brute-force index that searches the nearest neighbors using L2 distance. For example, in our case, an embedding vector representing an unsuccessful shopping session is queried against the index containing embeddings of successful sessions to find the $k$ nearest neighbors with the "successful" class label.

The nearest unlike neighbors retrieved represent instances that lie on the opposite side of the decision boundary. Each of these neighbors has different values for features such as **PRICE**, **SHPNG_COST**, and **PAYMNT_TYPE**, which makes them candidates for generating counterfactuals.

The overall process involves building the FAISS index, adding all dataset embeddings, associating each embedding with a class label, and then retrieving the $k$ nearest unlike neighbors using FAISS.

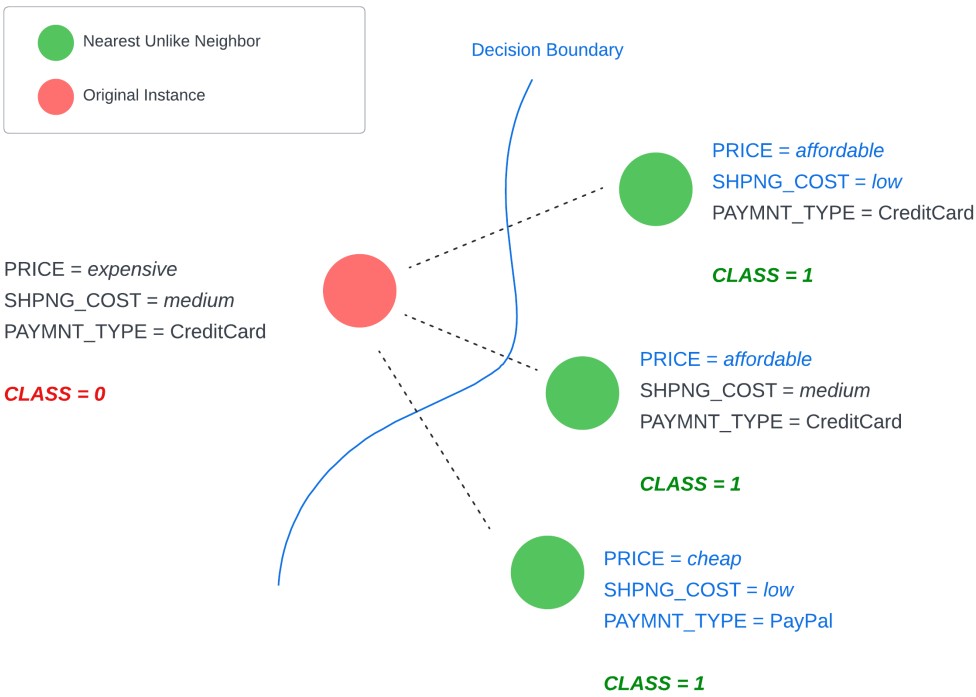

Figure 1: The original instance (red) has a **PRICE** of *expensive*, a **SHPNG_COST** of *medium*, and uses **PAYMNT_TYPE** = *CreditCard*. The class is **0**, indicating that this customer is not predicted to make a purchase. The nearest unlike neighbors (green) are counterfactual instances with slightly different features, located on the opposite side of the decision boundary (i.e., **CLASS = 1**, which are predicted to make a purchase). The nearest unlike neighbors have variations in **PRICE**, **SHPNG_COST**, and **PAYMNT_TYPE**.

Figure 1 illustrates the concept of nearest unlike neighbors in relation to the decision boundary. The original instance (red dot) is positioned near several nearest unlike neighbors (green dots) that lie on the opposite side of the decision boundary. This proximity allows for generating counterfactuals by modifying features in a way that crosses the decision boundary, achieving a different classification.

## 5.3 COUNTERFACTUAL SEARCH

Once we identify the nearest unlike neighbors, the next step is to generate counterfactuals using a greedy heuristic search method. In counterfactual search, we find data points that can be used to alter a classifier's decision, while making sure that they have a low distance from the original instance.

Using the neighbors identified in the previous section, the greedy heuristic search modifies features one at a time. Referring to Figure 1, the original instance has a **PRICE** of *expensive*, a **SHPNG_COST** of *medium*, and uses **PAYMNT_TYPE** = *CreditCard*. We modify each of these features individually to match one of the nearest unlike neighbors. For example, lowering the **PRICE** from *expensive* to *affordable* or lowering the **SHPNG_COST** from *medium* to *low* could flip the prediction from class 0 to class 1.

As the features are adjusted, the heuristic search keeps track of how each modification affects the classifier's decision. The goal is to generate the closest counterfactual, i.e., the one that modifies the fewest features while successfully flipping the class. In some cases, a combination of modifications, like changing both **PRICE** and **SHPNG_COST**, may be required.

The Counterfactual Generation Algorithm 1 seeks to identify the closest counterfactual instance to an original data point by iteratively modifying its features to achieve a desired target classification. Initiated with the original instance ($x_{\text{orig}}$), the algorithm progresses through the feature space by comparing against a set of nearest unlike neighbors ($\mathbb{N}$). For each neighbor, the algorithm creates $X_{\text{mod}}$, which is a set of new instances derived from $x_{\text{orig}}$ where each instance in $X_{\text{mod}}$ is generated by altering one specific mutable feature of $x_{\text{orig}}$ to match the corresponding feature in the neighbor.

Each modified instance, denoted as $x'$, is evaluated for its potential as a counterfactual: the modifications are checked to ensure they not only bring $x'$ closer to achieving the target classification but also maintain minimal distance from $x_{\text{orig}}$. This iterative process continues until a satisfactory counterfactual that meets the target classification is found, or all possibilities are exhausted. This method ensures that each proposed counterfactual is a minimal and interpretable adjustment to the instance that alters the model's decision.

## 6 EXPERIMENTS

We conducted experiments on a randomly held-out test set of 1,000 data points using all the counterfactual techniques in Section 4 and compared them using the following metrics:

- *Reconstruction Error:* Measures how closely a counterfactual instance resembles real-world data. It is calculated as the L2 norm between the autoencoder output and the counterfactual instance. A lower reconstruction error indicates a more plausible and realistic counterfactual Looveren & Klaise (2019); Dhurandhar et al. (2018); Nemirovsky et al. (2021).

$$E = \|AE(x_{cf}) - x_{cf}\|_2^2 \tag{1}$$

  where $AE$ is the autoencoder model and $x_{cf}$ is the counterfactual instance.

- *L1 Distance:* Measures the distance between the original and counterfactual instances. A lower L1 distance makes the counterfactual more actionable.

$$L1(x, x_{cf}) = \sum_i |x_i - x_{cf,i}| \tag{2}$$

  where $x$ is the original instance and $x_{cf}$ is the counterfactual.

- *Latency:* Represents the expected time to generate a single counterfactual. It is crucial for evaluating the practical usability of a counterfactual generation method.

---

**Algorithm 1** Counterfactual Generation

---

**Require:**
    $M$: Prediction model
    $E$: Encoders for categorical features
    $S$: Scaler for numerical features
    $\boldsymbol{x}_{\text{orig}}$: Original instance
    $\mathbb{N}$: Set of nearest unlike neighbors
    $T_{\max}$: Maximum iterations

    **Initialize:**
    $\boldsymbol{x}_{\text{cf}} \leftarrow \boldsymbol{x}_{\text{orig}}$                                 ▷ Initialize counterfactual candidate
    $d_{\min} \leftarrow \infty$                                      ▷ Initialize minimum distance
**Ensure:**
    $\boldsymbol{x}^*$: Optimal counterfactual instance
    $d_{\min}$: Minimum distance
  1: **for** $n \in \mathbb{N}$ **do**
  2:     **for** $t \leftarrow 1$ to $T_{\max}$ **do**
  3:         $X_{\text{mod}} \leftarrow \{\boldsymbol{x}' : \boldsymbol{x}'$ varies from $\boldsymbol{x}_{\text{cf}}$ by one feature towards $n\}$   ▷ Generate new instances with one modified feature
  4:         $(\boldsymbol{x}', d) \leftarrow \text{EvaluateModifications}(X_{\text{mod}}, \boldsymbol{x}_{\text{cf}}, E, S, M)$         ▷ Evaluate potential counterfactuals
  5:         **if** $d < d_{\min}$ **then**
  6:             $d_{\min} \leftarrow d$
  7:             $\boldsymbol{x}^* \leftarrow \boldsymbol{x}'$                              ▷ Update optimal counterfactual
  8:             $\boldsymbol{x}_{\text{cf}} \leftarrow \boldsymbol{x}'$                          ▷ Update current counterfactual
  9:         **end if**
10:         **if** $M(\boldsymbol{x}_{\text{cf}}) = $ target class **then**
11:             **return** $(\boldsymbol{x}_{\text{cf}}, d_{\min})$             ▷ Return if target class is achieved
12:         **end if**
13:     **end for**
14: **end for**
15: **return** $(\boldsymbol{x}^*, d_{\min})$

---

Table 2: A comparison of the five different counterfactual methods on four evaluation metrics. Note that the $\pm$ values represent the 95% confidence interval, calculated as $(1.96 \times \frac{\sigma}{\sqrt{n}})$, where $\sigma$ is the standard deviation and $n$ is the sample size.

| METHOD | % COVERAGE | RECONSTRUCT ERROR | L1 DIST | LATENCY (S) |
|---|---|---|---|---|
| NICE | **100.0** | $0.116 \pm 0.004$ | **1.79** | $1.641 \pm 0.055$ |
| DiCE | 95 | $0.364 \pm 0.054$ | 34 | $1.359 \pm 0.029$ |
| Standard GAN | **100.0** | $0.584 \pm 0.001$ | 53.46 | $0.012 \pm 0.002$ |
| CounteRGAN | **100.0** | $0.499 \pm 0.002$ | 52.83 | **$0.010 \pm 0.001$** |
| CFRL | 71.8 | $0.156 \pm 0.001$ | 32.47 | $0.145 \pm 0.002$ |
| Our method | **100.0** | **$0.112 \pm 0.003$** | 2.23 | $0.49 \pm 0.003$ |

Table 3: Coverage (%) for smaller, mutable feature sets.

| NO. OF FEATURES ($m$) | CFRL | DICE | OUR METHOD |
|---|---|---|---|
| 1 | 4 | 6.38 | **8** |
| 3 | 16 | 15.5 | **18.2** |
| 5 | 41.2 | 51.3 | **52.3** |
| 7 | 58.7 | **61.7** | 60.5 |

- *Coverage:* Represents the proportion of instances for which counterfactuals can be successfully generated.

$$C = \frac{\sum_{i=1}^{n} \mathbf{1}\left(M(x_{cf}^{(i)}) = d\right)}{n} \times 100 \tag{3}$$

where $n$ is the total number of instances, $M$ is the model, $d$ is the desired class (1 in our case), and $\mathbf{1}(\cdot)$ is the indicator function that equals 1 if its argument is true and 0 otherwise. It calculates the percentage of instances where the model's prediction for the counterfactual is the desired class.

As shown in Table 2, our method achieved the lowest reconstruction error and maintained 100% coverage. Although Generative Adversarial Networks (GANs) achieved the lowest latency and 100% coverage, they have large distance between the original and counterfactual instances. In addition, NICE has the lowest L1 distance, but does not allow mutability, which limits its usability, and is almost three times slower.

While Counterfactuals with Reinforcement Learning (CFRL) and DiCE allow customization of mutable features, they suffer from lower coverage, and significantly higher L1 distances and plausibility compared to our method. With this comparison, we can see that, in addition to better coverage and lower reconstruction errors, our method maintains a balance across all key metrics, including latency and distance, and also allows mutable features to be specified as desired. These experiments demonstrate that our method is not only strong in theory, but also works well in practice, making it very effective for real-world counterfactual reasoning.

To further examine this approach, we tested it on other mutable feature groupings. Even when we restrict mutability to smaller feature sets, such as [PRICE, SHPNG_COST, PAYMNT_TYPE] or only [PRICE], our method still outperforms CFRL and DiCE in almost every given configuration, as shown in Table 3.

$k$, representing the number of nearest unlike neighbors, acts as a hyperparameter affecting coverage and latency. A higher value of $k$ increases coverage but also increases the latency. To maintain reasonable latency, we set $k = 12$ to optimize for coverage.

The graphs in Figure 2 show how $k$ affects the coverage (Figure 2a) and latency (Figure 2b) of counterfactual generation. Coverage and latency both increase as $k$ increases, highlighting the trade-off

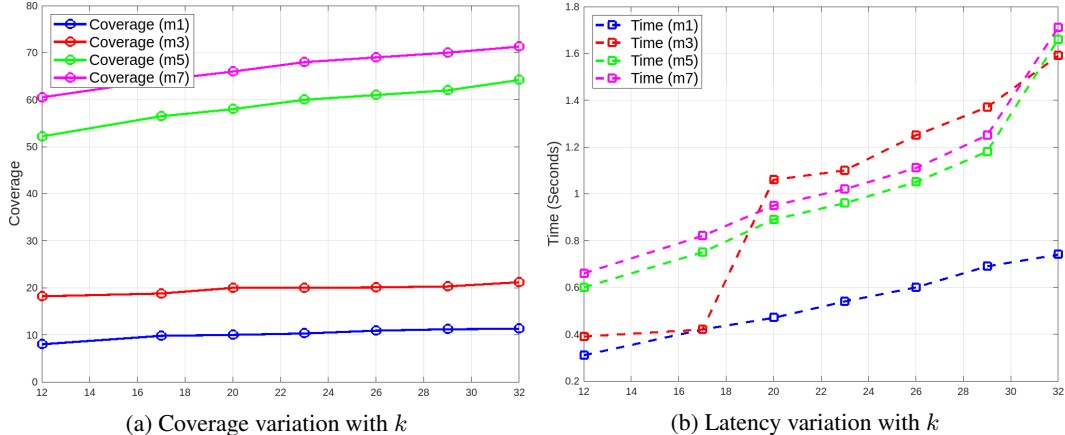

(a) Coverage variation with $k$                          (b) Latency variation with $k$

Figure 2: Comparison of Coverage and Latency variations with $k$

between the two metrics. Coverage with only one ($m1$) or three ($m3$) mutable feature is almost 2x lower than the scenarios with five ($m5$) or seven ($m7$) mutable features. Therefore, reducing the number of mutable features also reduces the model's ability to cover a wider range of counterfactuals.

The latency graph (Figure 2b) shows that as $k$ increases, the time required to generate counterfactuals also increases for all scenarios, which shows a trade-off between latency and the scope of features that can be modified. Therefore, as $k$ increases, the search for plausible counterfactuals becomes more thorough; however, it is also important to limit the search space in order to maintain a reasonably low computation time for deployment.

## 7 CONCLUSION

In this study, we introduced a novel counterfactual reasoning approach that uses embeddings generated by an eBERT model to create more accurate and actionable counterfactuals. We extend NICE's basic concepts by adding mutability and advanced natural language embedding techniques. This approach is especially effective in e-commerce settings where companies' fine-tuned embedding models capture domain-specific data relationships, thus increasing the plausibility of the counterfactuals.

Our experiments conducted on 200K shopping sessions show that our method outperforms existing counterfactual generation methods such as DiCE, GANs, and CFRL in terms of coverage, reconstruction error, and L1 distance, while maintaining a lower latency. The final latency of just 0.49 seconds per counterfactual indicates that our method is suitable for real-time applications.

Additionally, when testing the method with a limited set of mutable features, it consistently outperformed DiCE and CFRL. The parameter $k$, representing the number of nearest unlike neighbors, proved to be an imortant hyperparameter influencing both coverage and latency. Adjusting $k$ allows us to balance coverage against computational speed when searching for plausible counterfactuals, highlighting the flexibility of our approach. In future work, we will add functionality to specify a range of values for each feature, which will improve the customization of our method.

The results demonstrate that our approach not only enhances plausibility but also provides the best coverage and outperforms other methods offering mutability, specifically DiCE and CFRL, across all evaluation metrics.

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
