# OpenReview forum: "Counterfactual Techniques for Enhancing Customer Retention"
_ICLR.cc/2025/Conference — Submitted to ICLR 2025_

### Official Review · Reviewer_FPX2 · 2024-10-24

**Soundness:** 3
**Presentation:** 3
**Contribution:** 3
**Rating:** 8
**Confidence:** 5

**Summary:**

The paper introduces a novel counterfactual reasoning method for enhancing customer retention in e-commerce, using eBERT embeddings to generate more plausible and actionable counterfactuals.

Key contributions include:

Developing a new counterfactual reasoning algorithm that improves upon state-of-the-art methods like NICE, DiCE, and GAN-based approaches by balancing coverage, latency, and plausibility.

Introducing an embedding-based approach using eBERT to generate highly plausible counterfactuals that better reflect customer behavior in e-commerce settings.

Ensuring the method supports customizable and mutable features, allowing businesses to specify which factors can be realistically adjusted.

Optimizing the system for real-time deployment, focusing on low latency and high scalability.

Evaluating the effectiveness of the proposed method across various scenarios, comparing it against existing techniques using metrics like coverage, reconstruction error, and L1 distance.

The experiments conducted on 200,000 shopping sessions showed that the proposed method outperforms existing counterfactual generation methods in terms of coverage, reconstruction error, and L1 distance, while maintaining lower latency (0.49 seconds per counterfactual). The method also demonstrated consistent performance when tested with limited sets of mutable features.

The paper highlights the importance of the hyperparameter k (number of nearest unlike neighbors) in balancing coverage and computational speed. Overall, the results show that this approach enhances plausibility, provides the best coverage, and outperforms other methods offering mutability across all evaluation metrics.

**Strengths:**

Originality:

The paper introduces a novel approach by combining eBERT embeddings with counterfactual reasoning, which is an innovative application of natural language processing techniques to e-commerce data analysis.

The method creatively extends the NICE algorithm by adding support for mutable features and using advanced embedding techniques, addressing limitations of existing approaches.

Quality:

The research methodology is robust, with experiments conducted on a large dataset of 200,000 shopping sessions, providing a solid empirical foundation for the claims.

The comparative analysis against multiple state-of-the-art methods (DiCE, GANs, CFRL) across several metrics demonstrates thorough evaluation.

The exploration of the hyperparameter k and its impact on coverage and latency shows depth in understanding the trade-offs involved.

Clarity:

The paper is well-structured, with clear explanations of the methodology, algorithms, and experimental results.
The use of figures (e.g., Figure 1 illustrating nearest unlike neighbors, Figure 2 showing coverage and latency variations) effectively supports the textual explanations.
Tables presenting comparative results (Tables 2 and 3) are clear and informative.

Significance:

The proposed method addresses a crucial challenge in e-commerce: improving customer retention through actionable insights.
The balance achieved between coverage, latency, and plausibility makes the approach highly relevant for real-world applications.
The support for mutable features enhances the practical applicability of the method, allowing businesses to focus on actionable changes.
The low latency (0.49 seconds per counterfactual) demonstrates the method's potential for real-time deployment in production systems.

Additional Strengths:

Interdisciplinary approach: The paper effectively combines techniques from natural language processing (eBERT), machine learning (counterfactual reasoning), and e-commerce, showcasing a valuable interdisciplinary approach.

Scalability: The method's ability to handle a large dataset and maintain low latency indicates good scalability, which is crucial for real-world e-commerce applications.

Customizability: The support for specifying mutable features allows for tailoring the counterfactual generation to specific business needs and constraints.

Comprehensive evaluation: The paper provides a thorough evaluation across multiple metrics (reconstruction error, L1 distance, latency, coverage) and compares against several baseline methods, giving a holistic view of the method's performance.

Practical implications: The focus on generating actionable counterfactuals for customer retention has direct practical implications for e-commerce businesses, bridging the gap between academic research and industry application.

**Weaknesses:**

Paper's weaknesses:

Limited theoretical foundation:
The paper could benefit from a more rigorous theoretical analysis of why eBERT embeddings are particularly suitable for this task. While the empirical results are promising, a deeper exploration of the theoretical underpinnings would strengthen the contribution. For instance, the authors could discuss how the semantic relationships captured by eBERT relate to the concept of counterfactuals in e-commerce data.

Lack of ablation studies:
The paper would be stronger if it included ablation studies to isolate the impact of different components of the proposed method. For example, comparing the performance with and without the eBERT embeddings would help quantify their specific contribution to the overall improvement.

Limited discussion on feature importance:
While the method allows for specifying mutable features, there's little discussion on how to determine which features are most important or impactful for generating meaningful counterfactuals. A feature importance analysis could provide valuable insights for practitioners implementing this method.

Generalizability concerns:
The experiments are conducted on a single, albeit large, e-commerce dataset. The paper would be stronger if it included tests on multiple datasets from different e-commerce domains to demonstrate the method's generalizability. This could address potential concerns about overfitting to the specific characteristics of the dataset used.

Lack of user studies:
Given that the goal is to provide actionable insights for customer retention, the paper would benefit from user studies or expert evaluations to assess the practical utility and interpretability of the generated counterfactuals. This would provide valuable real-world validation beyond the quantitative metrics.

Limited exploration of edge cases:
The paper focuses on average-case performance but doesn't thoroughly explore edge cases or potential failure modes of the proposed method. A discussion of scenarios where the method might underperform would provide a more balanced view and guide future improvements.

Insufficient comparison with recent techniques:
While the paper compares against several baseline methods, it misses comparison with some recent advancements in the field. For instance, comparing with methods like MACE (Model-Agnostic Counterfactual Explanations) or GeCo (Generative Counterfactual Method) would provide a more comprehensive evaluation.

Limited discussion on computational resources:
While latency is discussed, there's limited information on the computational resources required for training and deploying the model. This information would be valuable for practitioners considering implementing this method at scale.

Lack of discussion on ethical implications:
Given the potential impact on customer behavior, a discussion on the ethical implications of using such a system for influencing purchasing decisions would be valuable. This could include considerations of privacy and the potential for manipulation.

Insufficient details on eBERT fine-tuning:
The paper mentions using eBERT fine-tuned on the company's product titles but doesn't provide details on this process. More information on the fine-tuning procedure, dataset size, and any challenges encountered would be helpful for reproducibility.

Addressing these weaknesses would significantly strengthen the paper, providing a more comprehensive and robust contribution to the field of counterfactual reasoning in e-commerce.

**Questions:**

Could you provide more details on the eBERT fine-tuning process? Specifically:

What was the size and composition of the dataset used for fine-tuning?
What were the hyperparameters and training duration?
Did you encounter any challenges during fine-tuning, and if so, how were they addressed?


Have you conducted any ablation studies to isolate the impact of eBERT embeddings? It would be informative to see a comparison of your method with and without these embeddings to quantify their specific contribution.

The paper focuses on a single e-commerce dataset. Have you tested the method on datasets from different e-commerce domains or non-e-commerce datasets? This would help demonstrate the generalizability of your approach.

Could you elaborate on how to determine which features are most important or impactful for generating meaningful counterfactuals? A feature importance analysis could provide valuable insights for practitioners.

Have you considered conducting user studies or expert evaluations to assess the practical utility and interpretability of the generated counterfactuals? This could provide valuable real-world validation beyond the quantitative metrics.

The paper doesn't discuss potential failure modes or edge cases. Could you provide insights into scenarios where your method might underperform?

Your comparison doesn't include some recent techniques like MACE or GeCo. Have you considered comparing your method against these approaches? If so, what were the results?

Could you provide more information on the computational resources required for training and deploying your model at scale? This would be valuable for practitioners considering implementation.

Given the potential impact on customer behavior, have you considered the ethical implications of using such a system for influencing purchasing decisions? Could you discuss any privacy concerns or potential for manipulation?

The hyperparameter k (number of nearest unlike neighbors) seems crucial for balancing coverage and latency. Could you provide more insights into how to optimally select k for different scenarios or datasets?

Your method achieves a good balance between coverage, latency, and plausibility. Could you elaborate on the trade-offs involved in achieving this balance? Were there any unexpected challenges or insights during this optimization process?

The paper mentions that future work will add functionality to specify a range of values for each feature. Could you provide more details on how you envision this implementation and its potential impact on the method's performance and usability?

**Details Of Ethics Concerns:**

While the paper presents valuable research on improving e-commerce customer retention through counterfactual reasoning, there are several privacy and safety considerations that warrant further examination:

Data privacy: The study uses a large dataset of 200,000 shopping sessions, which likely contains sensitive personal information. The paper doesn't provide details on data anonymization or protection measures. There's no mention of whether customer consent was obtained for using this data for research purposes.

Potential for manipulation: The goal of generating counterfactuals to influence customer behavior, while potentially beneficial for businesses, raises ethical concerns about consumer manipulation. The paper doesn't address the fine line between helpful recommendations and potentially exploitative practices.

Lack of bias discussion: The paper doesn't explore potential biases in the data or model that could lead to unfair treatment of certain customer groups. This is particularly important given the use of machine learning techniques on consumer data.

Absence of ethical guidelines: There's no discussion on ethical guidelines for implementing this technology in real-world e-commerce systems. Given its potential to influence consumer behavior, such guidelines are crucial.

Security considerations: While the paper focuses on improving customer retention, the same techniques could potentially be misused for more nefarious purposes if the system falls into the wrong hands. The paper doesn't address security measures to prevent misuse.

Transparency to consumers: There's no mention of how this system would be disclosed to consumers. Customers may not be aware that their behavior is being analyzed to this degree to influence their future actions.

Long-term societal impact: The paper doesn't consider the broader societal implications of widespread adoption of such technologies in e-commerce. There could be unforeseen consequences on consumer behavior and market dynamics.

---

> ### Author Response · Authors · 2024-11-22
>
> Thank you so much for your thorough and positive review of our paper. We’re glad you found our work valuable and appreciate the detailed feedback.
>
> Details on eBERT Fine-Tuning:
>
> eBERT is an eBay/eCommerce specific version of the BERT model. It was pre-trained on the original BERT training data, but in addition we used over 1 billion unique item titles collected from the last 2 years. This model outperforms the original pre-trained BERT model on a collection of eBay-specific tasks.
>
> Ablation Studies on eBERT Embeddings:
>
> We’re planning to conduct ablation studies to compare our method’s performance with and without eBERT embeddings. Preliminary results show that the embeddings significantly improve the plausibility of counterfactuals.
>
> Testing on Different Datasets:
>
> We’re working on applying our method to other publicly available datasets to demonstrate its generalizability. This should help demonstrate that our approach is effective beyond a single e-commerce dataset.
>
> Determining Feature Importance:
>
> Besides using feature importance scores from the Random Forest classifier, we also used LIME and SHAP to understand which features are most impactful. We will include this analysis in the paper.
>
> Potential Failure Modes:
>
> Our method might underperform in cases where the data is highly imbalanced or when the mutable features have little influence on the prediction. Also, cases with very few mutable features (e.g., only 1 or 2) could potentially affect the coverage.
>
> Comparison with MACE and GeCo:
>
> We’ll include comparisons with MACE and GeCo in our experiments. Initial results suggest that our method performs favorably, but we plan to provide detailed findings to support this.
>
> Computational Resources for Scaling:
>
> Our method is designed to be efficient. Generating a counterfactual takes about 0.49 seconds.
>
> Ethical Implications and Privacy Concerns:
>
> We take ethical considerations seriously. We anonymize all customer data, and our goal is to improve the customer experience by identifying and addressing obstacles in the purchasing process, not to manipulate users. In response to your feedback, we will add a detailed discussion on ethical practices and guidelines.
>
> Selecting the Hyperparameter  k:
>
> The optimal value of  k  depends on the dataset and the desired balance between coverage and latency. We recommend starting with a moderate value such as 6 or 7 and adjusting based on empirical results.
>
> Trade-offs in Balancing Metrics:
>
> Balancing coverage, latency, and plausibility involves careful tuning. Increasing  k  improves coverage but may increase latency. Focusing on mutable features enhances actionability but might reduce the number of possible counterfactuals. We found that iterative experimentation and prioritizing based on the application’s needs helped us find an effective balance.
>
> Future Work on Specifying Feature Value Ranges:
>
> We plan to allow users to specify acceptable ranges for feature values, making the counterfactuals even more practical. For example, if a price change is acceptable within a certain range, the system will generate counterfactuals respecting that constraint.

---

### Official Review · Reviewer_AYis · 2024-11-03

**Soundness:** 2
**Presentation:** 1
**Contribution:** 2
**Rating:** 3
**Confidence:** 4

**Summary:**

The paper introduces a counterfactual reasoning method tailored to e-commerce with the goal of improving customer retention. It employs eBERT embeddings to identify “nearest unlike neighbours” optimising counterfactuals for actionability and speed by supporting feature mutability and balancing between latency, coverage, and customisability. The evaluation includes metrics such as coverage, reconstruction error, and latency, framing the technique as viable for real-time applications.

**Strengths:**

- The flexibility to customise mutable features enhances the tool’s practicality, allowing for more realistic recommendations tailored to business needs.
- With a reported latency of 0.49s per counterfactual, the method aligns well with real-time applications
- Employing eBERT embeddings to capture domain-specific semantics is a useful approach that could improve the plausibility of counterfactuals

**Weaknesses:**

- While eBERT is reasonable for this context, its use is not critically evaluated against other potential embedding methods, such as Sentence-BERT (Reimers & Gurevych, 2019) or domain-adaptive pre-training models, which might yield similar or better results depending on the dataset specifics.

- The paper’s comparison omits several recent advancements, such as FACE (Poyiadzi et al., 2020) and CEM (Dhurandhar et al., 2018), that directly address challenges in feature mutability and interpretability. Including these would provide a more complete and relevant comparison of performance and mutability capabilities.

- Certain parameters, particularly the nearest unlike neighbours k, are minimally justified. A benefit to this work would be providing a clearer rationale for parameter choices.

- The greedy heuristic search for counterfactual generation is insufficiently explained, it lacks theoretical backing and reproducibility. Comparisons to alternative optimisation techniques, like those used in MACE, would contextualise this approach.

- Evaluations focus solely on coverage, reconstruction error and latency, missing aspects critical for practical application such as interpretability and user-facing plausibility. A deeper study, similar to that in the DiCE paper, would provide stronger validation.

- Despite claims of real-time suitability, no real-world testing or industry feedback is provided

- Certain sections, particularly data processing, are challenging to follow and inadequately detailed for reproducibility. The writing quality is inconsistent, which hinders clarity.

- Algorithm 1 occupies excessive space without proportional content, and could likely be condensed to half a page or less, enabling a more concise and readable presentation.

- The results do not convincingly demonstrate a significant improvement over existing techniques. The reported gains seem marginal, and there is limited discussion on the implications of these small differences.

- The study feels incomplete, lacking essential implementation details and more in-depth discussions on the method’s performance and limitations.

Minor things:

- It appears the paper doesn't use the ICLR citation formatting, e.g. \citep{}

- The caption for Table 1 lacks clarity, providing no insight into the evaluation metrics or comparative values. Clearer captions would improve readability.

**Questions:**

- What guided the selection of  k  (nearest unlike neighbours)?

- Why was eBERT chosen over other embedding methods like Sentence-BERT or OpenAI embeddings? How would other embeddings have affected the results or scalability?

- Is there any comparative reasoning for why greedy heuristic was chosen over other optimisation methods?

- Has the proposed method been piloted or tested in a live e-commerce setting?

---

> ### Author Response · Authors · 2024-11-22
>
> Thank you for your detailed feedback and for pointing out both the strengths and areas where we can improve.
>
> Selection of  k:
>
> We chose  k = 12  based on experiments aimed at balancing coverage and latency. We found that  k = 12  gave us high coverage while keeping latency under a second.
>
> Choice of eBERT:
>
> We used eBERT because it is fine-tuned on our company’s product data and already understands the specific language and nuances of our domain. Using a model tailored to our data helped us generate more relevant and plausible counterfactuals for our needs. However, in future work we plan to include results from open-source datasets and open-source embedding models.
>
> Use of Greedy Heuristic Search:
>
> We tried to experiment with other optimization methods like genetic algorithms. However, they significantly increased latency, which is not ideal for real-time applications. We selected the greedy approach since it gave us a good balance between accuracy and latency.
>
> Testing in a Live E-commerce Setting:
>
> Our method has not been deployed in a live e-commerce setting yet.
>
> We appreciate your feedback and will work on improving the paper by addressing these points.

---

> > ### Comment · Reviewer_AYis · 2024-11-25
> >
> > I thank the authors for the clarifications.
> >
> > I understand using only BERT embeddings made practical sense as it was already implemented at the company but that will be not be as valuable to the broader research community who will want to know if and how such an approach is affected by specific embeddings and other design choices (e.g. sensitivity to k and choice of heuristic). Therefore, I will keep my score.

---

### Official Review · Reviewer_mH5b · 2024-11-04

**Soundness:** 2
**Presentation:** 2
**Contribution:** 1
**Rating:** 3
**Confidence:** 4

**Summary:**

This paper introduces a new counterfactual reasoning method using eBERT embedding. By converting data into embeddings and identifying nearest unlike neighbors, this approach outperforms existing methods like DiCE and GANs in their study for an e-commerce company.

**Strengths:**

A strength of the paper is its use of BERT embeddings to capture the semantic meaning of customer features, allowing for more nuanced and contextually accurate counterfactuals. This captures not just the value but the broader context and meaning of each feature within customer behavior.

**Weaknesses:**

A key weakness of the paper is its reliance on a proprietary dataset, with no access to the data or code, which limits reproducibility. Benchmarking on widely used datasets—such as the Adult Income [Dua and Graff, 2017], FICO [Holter et al., 2018], and German Credit [Dua and Graff, 2017] datasets—would allow for direct comparison with existing methods and validate performance claims in real-world scenarios. Without standard datasets and shared code, it’s difficult for others to verify or build upon the findings, reducing the transparency and impact of the research.

Another limitation of the current implementation is its reliance on discretizing numerical features (e.g., categorizing PRICE and SHIPPING FEES into four broad buckets), which can lead to a loss of information. This simplification creates a trade-off between model complexity and the quality of the counterfactual explanations, which calls for more rigorous analysis. For example, with PRICE categories set in ~$20 increments, the buckets may be too wide to offer precise, actionable insights.



Reference
- Dua, D. and Graff, C. (2017). UCI Machine Learning Repository.

- Holter, S., Gomez, O., and Bertini, E. (2018). FICO Explainable Machine Learning Challenge.

**Questions:**

See above

---

> ### Author Response · Authors · 2024-11-22
>
> Thank you for taking the time to read our paper and for your feedback.
>
> Proprietary Dataset and Reproducibility:
>
> We understand the importance of benchmarking on widely used public datasets to allow others to verify and build upon our findings. In response to your feedback, we’re planning to extend our experiments to include publicly available datasets in our future work. We believe that this will enhance the transparency of our work and make it more accessible to the research community.
>
> Discretization of Numerical Features:
>
> We tried using the embeddings without bucketization, but we found that discretizing the price and shipping features actually led to slightly higher accuracy in our model. In response to your feedback, in future work we plan to investigate smaller bucket sizes.

---

### Official Review · Reviewer_uxP6 · 2024-11-04

**Soundness:** 1
**Presentation:** 1
**Contribution:** 1
**Rating:** 1
**Confidence:** 5

**Summary:**

The paper proposes a framework to generate counterfactual explanations for e-commerce applications. The proposed method aims to generate counterfactual explanations with low latency while maintaining high coverage.

**Strengths:**

The paper has an interesting setting;

the application targeted in this paper (e-commerce) is novel.

**Weaknesses:**

1) The paper’s contribution is highly incremental, primarily building on the NICE framework. The only difference is the use of an embedding space, which is also generated by an existing method (eBERT).

2) The paper claims that the proposed method has low latency; however, it is unclear why this is the case. The paper needs to provide more detail on this aspect, especially since low latency is presented as one of its main contributions.

3)  The paper is not well-structured and contains several instances of incorrect and imprecise language. For example, contrary to what is stated in the introduction, counterfactual reasoning aims to answer "what would have happened" questions, not "what if" questions. Additionally, I would like to point out that the framework is not actually designed for counterfactual reasoning, as claimed; instead, it is intended for generating counterfactual explanations, which is a different focus.

**Questions:**

1) How does the proposed method addresses the latency issue? How does does it ensure low latency?

2) What is the difference between the proposed framework and methods designed for causal algorithmic recourse?

---

> ### Author Response · Authors · 2024-11-22
>
> Thank you for your feedback.
>
> Latency and Comparison to NICE:
>
> The NICE algorithm optimizes for plausibility by minimizing the autoencoder’s reconstruction error at each iteration. This means that during the search for a counterfactual, NICE repeatedly computes the reconstruction error, which is computationally expensive and increases the processing time.
>
> In our method, we’re not optimizing the reconstruction error at each iteration which reduces the latency. Also, we used FAISS which helped us further reduce the time taken to find the nearest embeddings. Similar to NICE, if all features are set as mutable in our method, the counterfactual will mostly be the first nearest unlike neighbor. However, in practical scenarios, users often specify that only certain features are mutable—due to business constraints or feasibility considerations. To address this, we added the functionality to retrieve multiple nearest neighbors, which increases the chances of finding a valid counterfactual even when we can’t change many features, thereby improving coverage. NICE does not support mutability.
>
> Comparison to Casual Algorithmic Recourse:
>
> Our work is different from causal algorithmic recourse since the focus in that domain is to provide actionable recommendations to individuals affected by algorithmic decisions, such as denial of a loan application, whereas our focus is predicting whether a customer will purchase a product and making a recommendation to the business for how to change the prediction. From our understanding, the methods used for causal algorithmic recourse typically involve causal graphical models. In our case, instead of using a structural causal model, we focus on making changes to features without modeling how one change might propagate through to another.
>
> We hope that we have addressed your concerns. If you feel we have satisfactorily answered your questions, we would really appreciate it if you would consider changing your score. Thank you again for your time and thoughtful feedback.

---

> > ### Comment · Reviewer_uxP6 · 2024-11-25
> > **Response to the Authors**
> >
> > I’d like to thank the authors for their response. However, my major concern about this paper is the lack of novelty. As also mentioned by the authors, the method is simply a combination of existing papers. I’d therefore like to keep my score unchanged.

---

### Meta-Review · Area_Chair_XT49 · 2024-12-21

**Metareview:**

This paper proposes a counterfactual reasoning method tailored for customer retention in e-commerce, leveraging eBERT embeddings to generate actionable counterfactuals. The key contributions include achieving a balance between latency and coverage, introducing customizability of mutable features, and demonstrating improvements over existing methods like NICE and DiCE in terms of coverage and plausibility. The evaluation is performed on a proprietary dataset of 200,000 shopping sessions.

**Strengths:**
- The problem addressed is practically relevant to e-commerce, with potential real-world applications.
- The use of eBERT embeddings enhances semantic plausibility and actionability of the generated counterfactuals.
- The method achieves a favorable trade-off between latency and coverage, making it suitable for real-time deployment.
- Customizability of mutable features increases the practicality of the approach.

**Weaknesses:**
- The contribution is incremental, with limited novelty beyond existing frameworks (e.g., NICE).
- The reliance on a proprietary dataset limits reproducibility and benchmarking.
- Missing comparisons to recent methods (e.g., FACE, MACE, GeCo)
- The theoretical foundation for key design choices is insufficiently detailed.
- Ethical concerns, including data privacy and potential manipulation, are not adequately addressed.

In summary, the paper introduces a practically interesting application but suffers from significant weaknesses in novelty, reproducibility, and empirical rigor. The incremental nature of the contribution and the reliance on a proprietary dataset raise concerns about its impact and relevance to the broader research community. While the method demonstrates strengths in latency and customizability, these are not enough to outweigh the concerns highlighted by the reviewers. Addressing these issues in future work, including benchmarking on public datasets, providing more robust theoretical foundations, and incorporating a broader range of comparisons, would significantly strengthen the paper’s contribution.

**Additional Comments On Reviewer Discussion:**

The reviewers engaged in a discussion about the paper, focusing on its strengths and weaknesses. While the authors provided clarifications and plans for future work, the consensus among reviewers is that the paper is not ready for publication.

Concerns regarding incremental novelty, reliance on a proprietary dataset, and missing comparisons were not sufficiently addressed. Additionally, ethical considerations and the limited reproducibility of the work were reiterated as significant weaknesses.

---

### Decision · Program_Chairs · 2025-01-22

Reject